# A *Saccharomyces cerevisiae* Fermentation Product (Olimond BB) Alters the Early Response after Influenza Vaccination in Racehorses

**DOI:** 10.3390/ani11092726

**Published:** 2021-09-18

**Authors:** Alexandra Lucassen, Christa Finkler-Schade, Hans-Joachim Schuberth

**Affiliations:** 1Institute of Immunology, University of Veterinary Medicine Foundation, 30559 Hannover, Germany; alexandra.lucassen@tiho-hannover.de; 2Schade & Partner, 27283 Verden, Germany; cs@schadeundpartner.de

**Keywords:** *Saccharomyces cerevisiae*, prebiotics, postbiotics, vaccination response, neutrophilic granulocytes, T cells, B cells, reticulocytes, influenza vaccination, horses

## Abstract

**Simple Summary:**

Vaccination is one of the most important prophylactic methods for the prevention of disease in veterinary medicine. The initial post-vaccination events are dominated by innate immune mechanisms leading to a coordinated activation of specific adaptive immune responses. Thus, antibody production and induction of specific T cells depend on early post-vaccination events, triggered by vaccine ingredients and guided by factors released from initially targeted tissues and cells. The aim of this study was to analyse whether feeding a pre- and postbiotic feed supplement modulates such early immune responses after vaccination. Horses were fed a supplement (Olimond BB) based on products of fermented yeasts (*Saccharomyces cerevisiae*) for 8 weeks and subsequently vaccinated against influenza. Circulating leukocyte counts and their subpopulations were determined before and 24 h after vaccination. In absence of vaccination-induced side effects, horses receiving the supplement differed from control horses in the composition of neutrophilic granulocytes, CD4^+^ cells, and reticulocytes. Thus, the pre- and post-biotic feed supplement modulated early innate immune mechanisms after vaccination.

**Abstract:**

*Saccharomyces cerevisiae* (*S. cerevisiae*) fermentation products (SCFP) are used in animal husbandry as pre- and postbiotic feed supplements. A variety of immunomodulatory effects are noted in many species. The purpose of this study was to test the hypothesis that horses fed with SCFP containing feed additive Olimond BB display a modulated early immune response after influenza vaccination. Six horses received Olimond BB pellets (OLI) and five horses were fed placebo pellets (PLA) for 56 days. On day 40 all horses were vaccinated with a recombinant influenza A/equi-2 vaccine. At the day of vaccination, the groups did not differ in the composition of leukocyte subpopulations and reticulocytes. Twenty-four hours after vaccination total leukocyte counts and numbers of CD4^+^ T-cells significantly increased in both groups. In PLA horses, the numbers of neutrophil granulocytes significantly increased and numbers of CD8^+^ T-cells decreased, whereas the numbers of these cell types remained unchanged in OLI horses. Only OLI horses displayed a significant increase in reticulocyte percentages after vaccination. The numbers of lymphocytes, monocytes, CD21^+^ B-cells, and serum amyloid A levels remained unaffected in both groups after vaccination. Sixteen days after vaccination, PLA and OLI horses differed significantly in their enhanced ELISA IgG titres against Newmarket and Florida Clade 1 influenza strains. The observed differences after vaccination suggest that feed supplementation with Olimond BB leads to modulated early immune responses after influenza vaccination, which may also affect the memory responses after booster vaccination.

## 1. Introduction

*Saccharomyces cerevisiae* (*S. cerevisiae*) fermentation products (SCFP) are considered as nutraceuticals. Many studies report immunomodulatory and health benefits for SCFP applications in monogastric and ruminant mammalian species as well as in chicken [1,2]. The mechanisms of action of SCFP are manifold and include direct interaction of pre- and postbiotic ingredients with local gut epithelial [3,4] and immune [5] cells, as well as the modulation of gut microbiota composition [6,7]. However, immunomodulatory effects of SCFP are not restricted to the gut.

In dogs, feeding of S. cerevisiae mannan oligosaccharides (5–6 g/day for 24 days) increased blood neutrophil and decreased blood lymphocyte concentrations [8]. SCFP supplementation (125–500 mg/day for 28 days [9]; 0.65% of daily feed for 14 days, [10]) resulted in lower total WBC counts of dogs) and pigs (0.2% of daily feed [11]; 12–15 g/day [12]).

In adult cows, SCFP supplementation (20–90 g/animal/day for up to 60 days) altered the phenotype of circulating neutrophilic granulocytes, their phagocytosing capacity, and the gene expression of circulating leukocytes [13,14]. In calves, SCFP supplementation (1–5 g/day/animal) modulated both systemic and mucosal immune responses, resulting in reduced lung pathology, and a reduced incidence of secondary bacterial infection [15]. In beef calves challenged intravenously with LPS, SCFP-fed animals (12 g/day for 21 days) showed a reduced acute phase response and lower serum TNF levels [16]. In humans, SCFP supplementation (500 mg/day for 12 weeks) reduced the mean severity of specific allergic rhinitis symptoms, increased salivary IgA levels [17] and decreased incidences of cold or flu-like symptoms [18].

In horses, information concerning systemic effects of SCFP supplementation is still scarce. Valigura et al. [19] reported a stress-mitigating effect following prolonged exercise in young horses. Compared to a control group, horses that were SCFP dietary supplemented (21 g/d) for eight weeks showed no increase in serum amyloid A (SAA) values 6 h after a standardized submaximal exercise test and their cortisol levels returned quicker to pre-stress levels than in the control group. Dietary SCFP supplementation (21 g/day for 84 days) also decreased synovial prostaglandin E2 levels after an experimental LPS injection into the carpal joint of 10 month old Quarter horses [20].

Effects of SCFP supplementation on vaccination responses are scarcely reported. Following vaccination with a *Mannheimia haemolytica* vaccine, SCFP-fed beef cattle responded with a reduced acute phase response when compared to a control group [21]. In poultry, feeding of dietary yeast fermentate decreased antibody titers after Newcastle disease virus vaccination [22].

The immunomodulatory effects of SCFP feeding in several species and the horse implies that animals fed with SCFP display an altered response towards infectious challenges and vaccinations. The aim of our study was to address this hypothesis in racehorses vaccinated against equine influenza. Hence, SCFP were fed for eight weeks before horses were vaccinated with an inactivated influenza vaccine. To analyse early and late responses we determined the composition of blood leukocytes within 24 h after vaccination and titres of influenza-specific antibodies 2 weeks after vaccination.

## 2. Materials and Methods

### 2.1. Animals

The study was conducted at a stud farm in North Rhine Westphalia, Germany. The experiment was approved by the state of North Rhine Westphalia, Germany in accordance with § 8 (1) of the Animal Protection Act in conjunction with § 33 of the Animal Protection Experimental Animal Regulations (File number: 81-02.04.2020.A177). Approval for participation in this study in form of a signed declaration was obtained from the owner of the horses prior to the study.

The study was conducted with 11 English thoroughbreds (10 mares, 1 gelding) aged 2 years, with a mean body weight of 445 ± 32 kg. All horses were kept in one facility in individual boxes (3.50 m × 3.50 m) and different training courses. They received high-quality hay and concentrate 3 times a day as well as water ad libitum at all times. The horses were allocated to two groups. One group (6 animals) received Olimond BB (OLI) pellets and the other group (5 animals) received placebo (PLA) pellets (Table 1). OLI and PLA pellets were fed daily (10 pellets/day) during the routine evening feeding for a total period of 41 days.

### 2.2. Blood Samples

Immediately before each blood sampling, the horses were clinically examined according to good veterinary practice. This included the examination of oral and eye mucous membranes, lymph nodes, and the cardiovascular system. Rectal body temperature was recorded daily at 6 a.m. using a digital thermometer (Microlife^®^ Vet-Temp Thermometer, Covetrus, Hamburg, Germany). At 14-day intervals (days 1, 14, and 28) and at days 40 and 41 (day of vaccination and 24 h after vaccination, see Section 2.7), blood was collected from the jugular vein by puncture with a 20G × 1 Vacuette^®^ Multiple Use Drawing Needle (Becton Dickinson, Franklin Lakes, NJ, USA). Heparinized blood was collected in BD Vacutainer^®^ Sodium Heparin Tubes (Medicalis Medizintechnologie, Hannover, Germany). Blood for serum extraction was collected in BD Vacutainer^®^ CAT Tubes (Medicalis Medizintechnologie, Hannover, Germany). All blood samples were collected at the same day hour and in the same individual order. Analysis of samples took place within 24 h at the Institute of Immunology, University of Veterinary Medicine Foundation, Hanover, Germany.

### 2.3. Leukocyte Preparation and Subpopulation Determination

Total blood leukocyte counts and quantifications of major cell subpopulation fractions (lymphoid cells, granulocytes, monocytes) among blood leukocytes were determined as described [23]. Using a flow cytometer (BD Accuri™ C6 Flow Cytometer, Becton Dickinson Inc., Holdrege, NE, USA), 20,000 events were acquired. Viable granulocytes, monocytes and lymphocytes were identified in forward (FSC)/side scatter (SSC) density plots according to their characteristic FSC/SSC profiles (Figure 1c). Fractions of cellular subpopulations among viable leukocytes were multiplied with the total leucocyte count to obtain total cell counts.

### 2.4. Flow Cytometric Deterination of Lymphocyte Subpopulations

Leukocytes (0.3 × 10^6^/well) were incubated with monoclonal antibodies (mAK) in 96-well round bottom plates (Corning™Costar™ 96 well cell Culture Cluster, Round Bot-tom) for 30 min on ice. The monoclonal antibodies were pre-assembled in two sets (30 µL each). Set 1 contained anti eqCD4-FITC (Bio-Rad, MCA1078F, IgG1 0.1 mg/mL, 1:100), anti-canine CD21-AlexaFluor^®^647 (cross-reacting with horses, BIO-RAD, MCA1781A647, IgG1 0.05 mg/mL, 1:200) and anti eqMHC-II-RPE (BIO-RAD, MCA1085PE, IgG1, 100 tests, 1:10). Set 2 contained eqCD4-FITC and eqCD8-RPE (BIO-RAD, MCA1080PE, IgG2a 100 tests/mL, 1:10). Set 3 included isotype controls (BIO-RAD, MCA1209F, IgG1-FITC; BIO-RAD, MCA929PE, IgG2a-PE) of equal concentrations to ensure that no nonspecific reaction of the applied monoclonal antibodies with equine leukocytes occurred.

After incubation, cells were washed twice with 200 µL PBS containing bovine serum albumin 5.0 g/L, sodium azide 0.1 g/L) (membrane immunofluorescence (MIF) buffer) for 4 min at 400× *g*. The final cell pellet was suspended in 100 µL MIF buffer containing 2 µg/mL propidium iodide and subjected to flow cytometric analysis after gating on lymphoid cells (Figure 1c). All analyses were performed with the Accuri™ C6 Flow Cytometer software. Fractions of lymphoid cells (CD4^+^ T-cells, CD8^+^ T-cells, CD21^+^ B-cells, and CD21^−^/MHC-II^+^ lymphoid cells) were multiplied with the absolute number of lymphoid cells/mL blood.

### 2.5. Flow Cytometric Deterination of Reticulocytes

Reticulocyte percentages were determined essentially according to the procedure described by Viana et al. [24]. In brief, heparinized blood (5 µL) was mixed with 2 mL PBS and 20 µL acridine orange solution (5 µg/mL PBS, Sigma-Aldrich, Taufkirchen, Germany). An unstained sample served as control. Samples were incubated at room temperature in the dark for 30 min. Subsequently, the fraction of reticulocytes among erythrocytes were determined flow cytometrically (Figure 1f–i).

### 2.6. Vaccination

On day 40, horses of both groups were vaccinated with a commercial vaccine against influenza (PROTEQFLU™ (1 mL Influenza A/eq/Ohio/03 [H3N8] recombinant of canary pox virus (strain vCP2242) and influenza A/eq/Richmond/1/07 [H3N8] recombinant of canary pox virus (strain vCP3011), with carbomer as the adjuvant, Boehringer-Ingelheim Vetmedica GmbH, Deutschland). The vaccination was administered intramuscularly on the left side of the neck with a disposable needle (1 × 20G, Neoject^®^, Dispomed Witt, Germany) after cleaning according to standard practice. In addition to rectal body temperature and a general examination, the animals were also examined 24 h after vaccination for vaccination side effects, such as pain and swelling of the injection site.

### 2.7. Determination of Serum-Amyloid A and Influenza-Specific Antibodies

For photometric analysis of the major acute-phase serum amyloid a (SAA) and determination of influenza-specific antibody titers, blood serum was collected on days 40 and 41. Samples were analyzed in an external laboratory (Laboklin, GmbH & Co. KG, Bad Kissingen, Germany). The ELISA was performed with equine influenza virus strains: A/eq/Prague/56, A/eq/South Africa/04/2003, A/eq/Ohio/03, A/eq/Newmarket/1/93, A/eq/Newmarket/2/93.

### 2.8. Statistical Analysis

Data are expressed as mean ± SEM and the number of subjects as n. Comparison of all parameters between the two time points in each group was performed using a paired *t*-test (SAS Enterprise Guide 7.1). In advance, the variables per group were tested for normal distribution using the Shapiro-Wilk, Kolmogorov-Smirnov, Cramer-von Mises, and Anderson-Darling tests. If there was no normal distribution in the OLI group with *n* = 6, the sign rank test was applied. If there was no normal distribution or log-transformed normal distribution in the PLA group with *n* = 5, a paired *t*-test was performed for descriptive purposes. The following parameters were not normally distributed in the PLA: Serum amyloid A, Florida clade 1 strain-specific and Newmarket strain-specific antibodies, reticulocytes, CD4^+^, CD8^+^, and CD21^+^ cells. Differences were considered statistically significant at a *p* value of < 0.05.

## 3. Results

### 3.1. Clinical Response and Serum Amyloid A Concentrations

At the day of vaccination, the general condition of all horses was normal and inconspicuous. Before vaccination, the rectal body temperature in both groups was within the physiological range (OLI: 37.86 ± 0.15 °C; PLA: 38.04 ± 0.05 °C). Horses of both groups showed no local vaccination side effects such as swelling, pain, or warmth at the injection site 24 h after vaccination. Twenty-four hours after vaccination the body temperature in both groups did not change significantly (OLI 37.84 ± 0.11 °C; PLA 38.06 ± 0.05 °C). Serum amyloid A concentrations showed a >2-fold increase in 2/5 and 1/6 horses of the PLA and OLI group, respectively (Table 2).

### 3.2. Blood Leukocyte Numbers and Reticulocyte Percentages

Before vaccination (day 40), both groups displayed the same numbers of leukocytes, leukocyte subpopulations, and reticulocytes except for a tendency (*p* < 0.1) for lower numbers of CD8^+^ T cells in the OLI group (Table 3).

Twenty-four hours after vaccination, the number of total leukocytes and CD4^+^ cells increased significantly in both groups (Figure 2a,f). In the PLA group neutrophil counts increased (*p* = 0.033) and CD8^+^ cells decreased (*p* = 0.031), whereas in the OLI group these cell populations remained unchanged (Figure 2c,g). A vaccination-induced rise in reticulocyte percentages was only apparent in the OLI group (*p* = < 0.005, Figure 2b). In neither group the vaccination altered the numbers of lymphocytes, monocytes, CD21+ cells or CD21^−^/MHCII^+^ cells (Figure 2d,e,h,i).

The observed changes in blood leukocyte and cell subpopulation numbers ranged between a 0.5-fold decrease (Figure 3, PLA, CD8^+^ T cells) and a 1.9-fold increase in CD4^+^ T-cells (Figure 3, OLI, CD4^+^ T cells).

### 3.3. Vaccinaton-Induced Influenza-Specific Antibodies

On the day of challenge (d40), horses of both groups had comparable ELISA antibody levels against H3/N8 influenza. Before and after vaccination antibodies against a H7/N7 strain were not detectable (Table 3). Sixteen days after vaccination horses of both groups showed a numerical increase in ELISA titers against H3/N8 strains. In the OLI group the increase reached significance for antibodies against Florida Clade 1 (H3/N8), in the PLA group a significant titer increase was noted for antibodies against Newmarket 2/93 (H3/N8) (Table 4).

## 4. Discussion

The period of six weeks in which feed was supplemented with SCFP did not result in an altered composition of circulating immune cells as at the day of vaccination both groups (PLA, OLI) did not differ significantly in any of the measured parameters (Table 2). This is in line with findings in calves and lambs where the SCPF feeding had no influence on the fraction of circulating immune cells subpopulations [15,25]. In contrast, according to the literature, SCFP supplementation raised the numbers of circulating leukocytes and neutrophils in heat-stressed cows [26] and the fraction of monocytes and B-cells in adult dogs [9]. These effects of SCFP supplementation on circulating immune cell populations indicate that SCFP feeding may not always lead to an altered blood immune cell composition and that species, age of the animals, feeding regimes, housing and other conditions affect the outcome of an SCFP supplementation. In the present study with racehorses, a changed leukocyte composition could not serve as an indicator of assessing a possible immunomodulatory effect induced through SCFP feeding.

Vaccinated horses in this study did not show any side effects after vaccination. None of the horses of either group displayed an enhanced (>38.5 °C) body temperature or swelling at the site of injection 24 h after vaccination. This may be due to the low number of vaccinated horses, since transient fever after vaccination against influenza was reported for several of 34 race horses when analysing the effects of 3 different influenza vaccines [27]. Concomitant with absent clinical side effects, serum amyloid A values remained largely unchanged after vaccination (Table 2). Thus, vaccinations induced only a weak or absent acute phase response (APR). Only two of five (PLA) and one of five (OLI) horses displayed a two-fold increase of SAA concentrations 24 h after vaccination (Table 2). This is in contrast to prominent acute phase responses with up to 100-fold higher SAA values as reported after vaccination with equine influenza vaccines containing ISCOMS (immune stimulating complexes) or a vector virus [28], or after booster vaccinations against EHV 1/4 within 24 h after injection [29]. The lack of an APR in the PLA group precludes speculations, whether SCFP supplementation in horses would lead to a reduced APR as reported for SCFP-fed beef steers following vaccination with a *Mannheimia haemolytica* vaccine [21].

Strong APRs after vaccination of horses were associated with an increased number of white blood cells between 9 and 48 h after vaccination [28]. Our observation of a vaccination-induced changed blood cell composition (Figure 2 and Figure 3) indicates that a strong APR does not seem to be a prerequisite for such changes. Therefore, although we did not determine serum cytokine concentrations, the observed changes in blood leukocyte composition are most likely not due to high concentrations of vaccination-induced inflammatory cytokines (e.g., IL-1β, IL-6, TNF). The effects reported in the racehorses may be due to a low level induction of cytokines, chemokines and other mediators, as observed after vaccination of humans against influenza [30] and more recently when analyzing the innate immune transcriptional signatures of peripheral blood mononuclear cells (PBMC) one day after vaccination of humans [31,32].

The differences between PLA and OLI horses regarding the vaccination-induced change in blood cell composition indicates that SCPF-fed horses release a different set of systemically acting mediators after vaccination. This is indicated by the selective and significant rise of reticulocytes after vaccinating OLI group horses (Figure 2). Raised reticulocyte fractions are the result of an increased erythropoiesis [33], an altered maturation of erythrocytes [34] or a differential activity of liver and spleen macrophages. An enhanced generation of reticulocytes after vaccination has/has not been described yet. Erythropoiesis normally takes place in the bone marrow [20,21]. Under specific conditions, e.g., stress situations or after vaccination, extramedullary erythropoiesis can also take place in the spleen and liver [35,36]. Whether vaccination of horses leads to the upregulated expression of genes encoding erythropoiesis-involved transcription factors and other erythroid proteins, or whether vaccination-induced mediators modulated the function of macrophages involved in erythrocyte turn-over [37] is unknown. In the present study, the observed differential changes in peripheral blood leukocyte composition may serve as a surrogate for immune responses taking place in the muscle at the site of vaccination. An altered tissue responsiveness after SCFP feeding has been reported for human bronchial epithelium [38], the bovine udder [13,39], the bovine uterus [40], and equine joints [20].

Early (18–48 h post vaccination) changes in PBMC gene expression involve a multitude of genes including those coding for chemokines, chemokine receptors, and adhesion molecules [41,42]. A putative altered leukocyte expression of these molecules after vaccination of horses could lead to enhanced emigration from blood into tissues, or an enhanced release from primary and secondary immune organs, thus explaining the altered leukocyte composition in peripheral blood. The observed fold-changes up to 1.9 (e.g., CD4^+^ T cells, Figure 3) are in line with findings in healthy human volunteers vaccinated intramuscularly with an adjuvanted swine flu vaccine. This resulted in an average 1.6-fold increase of the blood monocyte fraction, while whole blood cells numbers slightly decreased [42]. Interestingly, such early changes are very useful to predict post-vaccination (antibody) responses. This has been shown in humans vaccinated with seasonal and pandemic H1N1 (pH1N1) vaccines [43].

For all horses the vaccination represented a booster vaccination with the aim to raise the pool of circulating, influenza-specific antibodies. The booster response was detectable at day 14 post vaccination with a selective increase of antibody titers against H3N8 strains (Table 4). Interestingly, the increase of antibody titers against several influenza strains was significantly different between PLA and OLI horses (Table 4). Whether these differences in antibody titer increases also translate into different levels of protection against equine influenza still needs to be determined. The single radial hemolysis assay, offering a stronger correlation of antibody titers with protection [44], was not performed. At least, the observed differential antibody responses towards a booster vaccination and whether the differential changes in circulating leukocyte numbers and reticulocytes are indicative for a modulated memory response is still speculative. At least in humans, early vaccination-induced patterns of gene expression predicted antibody titers against influenza or correlated with the humoral immune response to influenza vaccination [45,46].

## 5. Conclusions

The different changes in blood leukocyte composition and reticulocyte fractions of PLA and OLI racehorses after an influenza booster vaccination indicate that feeding of *S. cerevisiae* fermentation products alters initial vaccination-induced responses including the release of systemically acting mediators. The SCFP-induced systemic immune modulation may not only affect vaccination-induced responses but also initial events after pathogen contact. Initial innate immune responses towards vaccines and pathogens are decisive for the induction of adaptive immune responses. As we only studied the response to a booster vaccination, the question remains, whether SCFP feeding also modulates the generation of memory T and B cell pools of naïve horses and how the feeding-induced immune modulation affects the response to different classes of vaccines. In sum, the observed immunomodulatory effects of SCFP in horses suggest that SCFP-fed horses may display an altered response towards infectious challenges and vaccinations.

## Figures and Tables

**Figure 1 animals-11-02726-f001:**
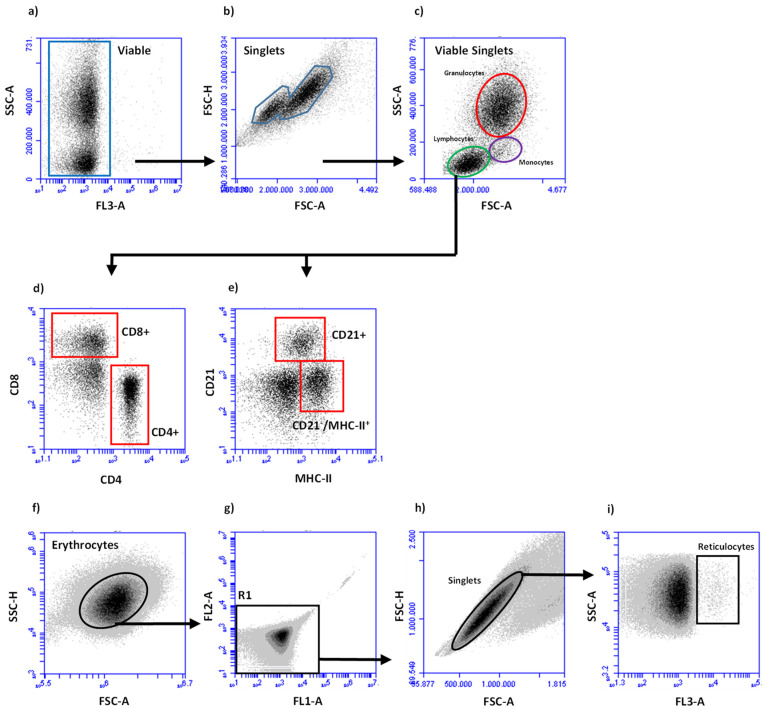
Gating strategy to determine fractions of leucocytes (**a**–**e**) and reticulocytes (**f**–**i**). (**a**) Viable, propidium iodide-negative cells were determined in FL3 versus SSC-A density plots.; (**b**) Single cells among viable cells were identified in FSC-A versus FSC-H density plots; (**c**) A region of granulocytes, lymphocytes, and monocytes were identified in FSC-A versus SSC-A density plots. Fractions of lymphoid cell subpopulations were determined after dual staining with antibodies specific for CD4 and CD8 or MHC II and CD21 to identify CD4^+^ and CD8^+^ lymphocytes (**d**) or CD21^+^ B-cells and CD21^−^MHC-II^+^ lymphocytes. Erythrocytes were identified in FSC/SSC density plots (**f**) and gated for FL1-/FL2- events (region R1, (**g**)). After gating on single cells (**h**), reticulocyte fractions were determined in FL3/SSC density plot (**i**).

**Figure 2 animals-11-02726-f002:**
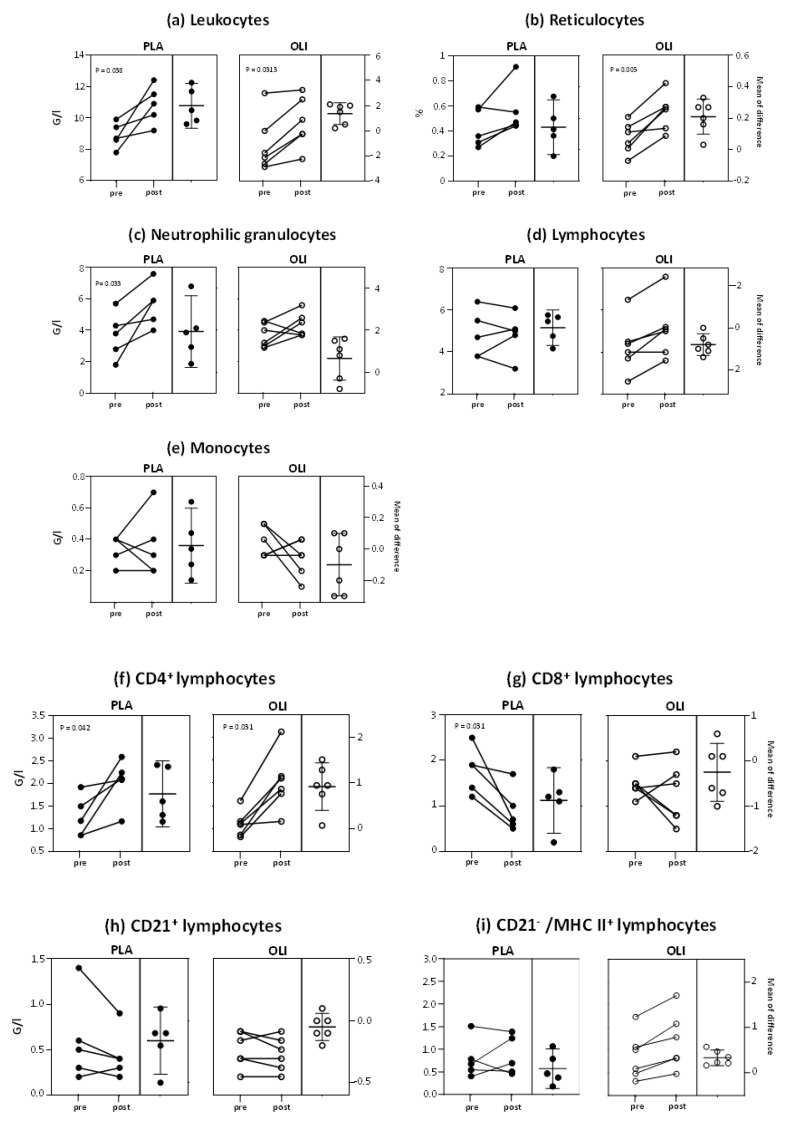
Vaccination-induced changes in leukocyte counts and reticulocyte proportions in blood. Cell counts and proportions of reticulocytes were determined by flow cytometry on the day of vaccination (pre) and 24 h after vaccination (post). Shown are individual responses and mean differences of vaccinated horses (OLI, *n* = 6; PLA, *n* = 5). *p* values were determined by paired *t*-test or signed-rank test and describe significances before and after vaccination within a group.

**Figure 3 animals-11-02726-f003:**
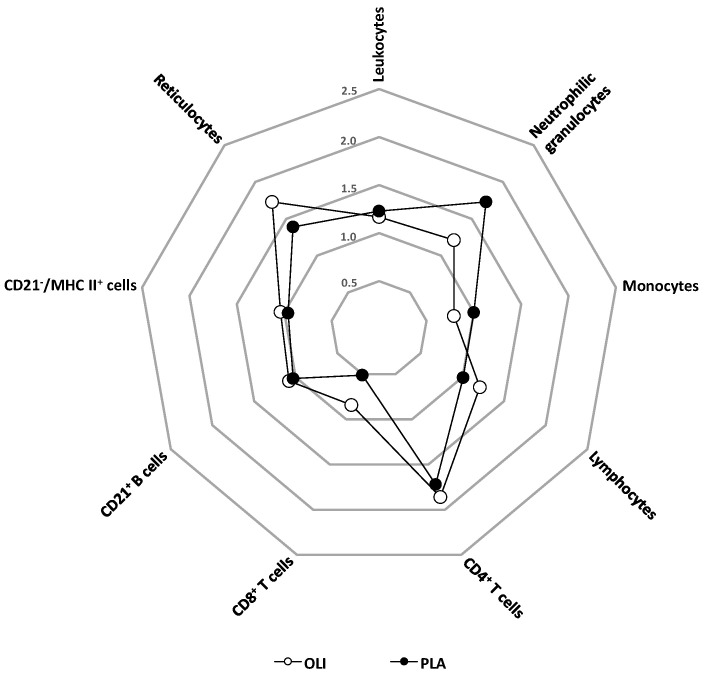
Spider diagram showing fold changes of blood cell populations within 24 h after vaccination.

**Table 1 animals-11-02726-t001:** Placebo and Olimond BB feed supplement ingredients.

Ingredient	PLA ^1^	OLI
Tocopherol extract	0.05 g	0.05 g
Coconut oil	0.15 g	0.15 g
Vitamin C	0.15 g	0.15 g
Dextrose	1.00 g	-
Corn cob meal	0.75 g	-
Linseed cake	0.75 g	-
Microcrystalline cellulose	1.03 g	1.43 g
Minerals	1.13 g	1.23 g
inactivated yeasts	-	2.00 g
Pellet	5.01 g	5.01

^1^ Feed supplements were fed as pellets: PLA, placebo; OLI, Olimond BB (g per pellet). Each horse received 10 pellets/day.

**Table 2 animals-11-02726-t002:** Serum Amyloid A concentrations before and 24 h after vaccination.

Group	Horse	SAA (µg/mL)		Fold Increase ^1^
		D40	D41	
OLI	#1	8.8	69.8	7.9
	#2	10.7	11.8	1.1
	#3	10.5	13.5	1.3
	#4	233.7	141.9	0.6
	#5	9.2	16.1	1.8
	#6	10.9	11.0	1.0
PLA	#7	9.7	31.0	3.2
	#8	9.5	11.8	1.2
	#9	11.0	14.8	1.3
	#10	10.4	11.4	1.1
	#11	10.9	58.9	5.4

^1^ ratio between day (D) 41/40 SAA values.

**Table 3 animals-11-02726-t003:** Concentrations of blood cellular subpopulations on the day of vaccination.

Cell Type	OLI	PLA	*p*
Leucocytes (G/L)	8.35 ± 1.79	8.88 ± 0.80	0.557
PMN (G/L)	3.69 ± 0.74	3.67 ± 1.45	0.971
Monocytes (G/L)	0.38 ± 0.09	0.36 ± 0.09	0.752
Lymphocytes (G/L)	4.28 ± 1.30	4.85 ± 1.12	0.460
CD8^+^ T cells (G/L)	1.51 ± 0.33	1.76 ± 0.51	0.086
CD4^+^ T cells (G/L)	1.11 ± 0.28	1.27 ± 0.45	0.471
CD21^+^ B cells (G/L)	0.49 ± 0.20	0.59 ± 0.47	0.633
CD21^−^/MHCII^+^ cells (G/L)	0.87 ± 0.52	0.78 ± 0.43	0.782
Reticulocytes (%)	0.34 ± 0.13	0.42 ± 0.15	0.371

Mean ± SEM of cell types after 40 days of OLI, Olimond BB (6 horses) or PLA, Placebo (5 horses) supplementation. *p*: *p*-value.

**Table 4 animals-11-02726-t004:** ELISA titer of Influenza-specific antibodies.

	Antigen ^1^							
Group	Florida Clade 1 (H3N8)			Newmarket (H3N8)			Prague (H7N7)	
	D40	D56	*p*	D40	D56	*p*	D40	D56
OLI	5.13 ± 0.62 ^a2^	5.96 ± 1.05 ^a^	0.033	4.71 ± 1.33 ^a^	5.27 ± 0.79 ^a^	0.373	n.d. ^3^	n.d.
PLA	5.41 ± 0.58 ^a^	6.24 ± 0.49 ^a^	0.108	4.30 ± 0.58 ^a^	5.68 ± 0.31 ^a^	0.003	n.d.	n.d.

^1^ strains used for the ELISA: Florida clade 1 (A/eq/South Africa 04/2003, A/eq/Ohio/03), Newmarket (A/eq/Newmarket/1/93, A/eq/Newmarket/2/93), Prague (A/eq/prague/56). ^2^ log-transformed (natural logarithm) titer means ± SEM. Group differences at days 40 and 56 were tested with the student’s *t*-test. Means in columns with the same small letter superscript indicate no significant statistical difference. Statistical differences between days 40 and 56 were determined with the paired student’s *t*-test. ^3^ not detectable.

## Data Availability

None of the data were deposited in an official repository. The data that support the study findings are available upon reasonable request.

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
