# Peer review of "A Saccharomyces cerevisiae Fermentation Product (Olimond BB) Alters the Early Response after Influenza Vaccination in Racehorses"

_animals, 2021, doi:10.3390/ani11092726_

Round 1
Reviewer 1 Report
Manuscript animals-1377706, entitled “A Saccharomyces cerevisiae fermentation product (Olimond BB) alters the early response after influenza vaccination in race-horses”
Recommendation: The above paper is not suitable for publication in its present form.
General comment
The article provides useful information about the effects of a Saccharomyces cerevisiae fermentation product (Olimond BB) on early immune response after influenza vaccination in race-horses. Although, the experiment is in general appropriately designed and implemented, there are some points that should be corrected or clarified.
My main concern is the small sample size. How many were the horses used? Eleven (L92) or ten (Table 2)?
Please replace “food” with “feed” throughout the text (L15, 16, 23, 96 etc)
In introduction, please provide references according to species. First dogs, then pigs, cows-calves and finally humans. At the same time, please also provide the levels of SCFP supplementation.
L43: “…products (SCFP) are considered as nutraceuticals…”
L44: “…for of SCFP applications in…”
L47: “…and immune [5] cells, as…”
L50: “…blood neutrophil and decreased blood lymphocyte levels [8]. In…”
L53: “dietary inclusion” instead of “feeding”
L58-59: “…and decreased incidences of cold…”
L64-65: “…horses that were SCFP dietary supplemented for eight weeks showed no…”
L72: What is the meaning of “(?)”
L81-83: Please delete
L86: “experimental” instead of “stud”
L87: Please delete “animal”
L96: “allocated” instead of “divided”
L96: Please delete “verum”
L98: “OLI” instead of “Verum”
L103: “sampling” instead of “collection”
L112: “collected” instead of “taken”
L120: “presented” instead of “given”
L160: “performed” instead of “done”
L166: “…by Viana et al. [23].”
L189: “are” instead of “were”
L196: “The following parameters…”
L223: “…percentages that was only…”
L242: “increase reached significance” Where are these data presented?
L252-254: Please remove this sentence to conclusions
L254-255: Pleas delete
L260: “In contrast, according to the literature, SCFP supplementation…”
L266: “an indicator of assessing a possible” instead of “a parameter for an”
L273-275: Please rephrase
L290: “reported” instead of “seen”
L297: “indicated” instead of “best seen”
L336-337: Please specify. Leucokyte for both groups, reticulocyte only for OLI group.
Author Response
Thank you very much for your careful review!
My main concern is the small sample size. How many were the horses used? Eleven (L92) or ten (Table 2)?
Due to our mistake, one horse of the OLI group was not listed in Table 2. This is now corrected. We have data for eleven vaccinated animals. The small sample size has been very carefully taken into account when performing the statistics (see 2.8. Lines 175ff).
Please replace “food” with “feed” throughout the text (L15, 16, 23, 96 etc)
done
In introduction, please provide references according to species. First dogs, then pigs, cows-calves and finally humans. At the same time, please also provide the levels of SCFP supplementation.
We restructured the respective lines and provide details about the SCFP supplementation (also in lines dealing with SCFP supplementation in horses)
L43: “…products (SCFP) are considered as nutraceuticals…”
done
L44: “…for of SCFP applications in…”
done
L47: “…and immune [5] cells, as…”
done
L50: “…blood neutrophil and decreased blood lymphocyte levels [8]. In…”
done
L53: “dietary inclusion” instead of “feeding”
done
L58-59: “…and decreased incidences of cold…”
done
L64-65: “…horses that were SCFP dietary supplemented for eight weeks showed no…”
Done
L72: What is the meaning of “(?)”
Sorry, this were remains of older internal revisions (now deleted)
L81-83: Please delete
done
L86: “experimental” instead of “stud”
Well, in fact this was not an experimental farm. Thus, we'd like to keep the term stud farm.
L87: Please delete “animal”
done
L96: “allocated” instead of “divided”
done
L96: Please delete “verum”
done
L98: “OLI” instead of “Verum”
done
L103: “sampling” instead of “collection”
done
L112: “collected” instead of “taken”
done
L120: “presented” instead of “given”
Based on the remarks of Reviewer 2, sections 2.3. and 2.4. are now combined and significantly shortened
L160: “performed” instead of “done”
done
L166: “…by Viana et al. [23].”
done
L189: “are” instead of “were”
done
L196: “The following parameters…”
done
L223: “…percentages that was only…”
To make the sentence more accurate and understandable we changed “vaccination induced” into “vaccination-induced”. Then “percentages was” makes more sense
L242: “increase reached significance” Where are these data presented?
Data are presented in Table 4 (in the text, we corrected the reference to Table 4 in Line 244)
L252-254: Please remove this sentence to conclusions
done
L254-255: Pleas delete
done
L260: “In contrast, according to the literature, SCFP supplementation…”
done
L266: “an indicator of assessing a possible” instead of “a parameter for an”
done
L273-275: Please rephrase
We changed the sentence into. “Concomitant with absent clinical side effects, serum amyloid A values remained largely unchanged after vaccination (Table 2). Thus, vaccinations induced only a weak or absent acute phase response (APR)”
L290: “reported” instead of “seen”
done
L297: “indicated” instead of “best seen”
done
L336-337: Please specify. Leucokyte for both groups, reticulocyte only for OLI group.
For a conclusion, we feel that a repetition of all details is not fully appropriate. In addition to leukocytes and reticulocytes we would also have to mention CD4+ and CD8+ T cells, as well as neutrophilic granulocytes. To indicate that the groups differed in vaccination-induced changes (the key finding) we changed in Line 340 “The changes” into “The different changes”. This indicates that both groups displayed vaccination-induced changes without a need to repeat the details.
In addition, we rephrased the sentence as follows: “The different changes in blood leukocyte composition and reticulocyte fractions of PLA and OLI racehorses after an influenza booster vaccination indicate that feeding of S. cerevisiae fermentation products alters initial vaccination-induced responses including the release of systemically acting mediators.”
Reviewer 2 Report
This manuscript by Lucassen and colleagues has the effects of a Saccharomyces cerevisiae fermentation product (Olimond 2 BB) on the early response after influenza vaccination in race horses. This study found that food supplementation with Olimond BB leads to modulated early immune responses after influenza vaccination, which may also affect the memory responses after booster vaccination. This is an interesting topic. The study was well designed. The data was well presented. The writing needs to be carefully checked and improved. The following changes could improve the quality of the paper.
- Line 17, what is the “Olimond BB”? Please provide the detailed information.
- Line 100, 1) the table tile need to be more specified, such as the name of the pellet; 2) please add the full name explanation of PLA and OLI in the table note; 3) what is the total amount for each pellet?
- Too much details for the material and methods. Please reduce them by cite the reference.
- Line 201, 1) please do not use the bond style for D40 and D41. Please check the similar issues throughout the paper; 2) why not use the mean ± SD to express the results?
- Line 218, please correct “P-value” to “P-value”.
- It is not the right way to express the “Olimond BB (OLI) or Placebo (PLA)” in the table note. It should be appeared as “OLI, Olimond BB; PLA, Placebo”.
- Line 233, please correct “(P = < 0.005)” to “P ≤ 0.005”.
- Line 229, please correct “(OLI, n =6; PLA, n =5)” to “(OLI, n = 6; PLA, n = 5)”.
- Line 2447, please correct “vaules ± standard deviation” to “means ± SEM”, which should consistent with the previous presentation.
- Please carefully check the reference and make them follow the right style of the journal.
Author Response
Thank you very much for your careful review!
1.Line 17, what is the “Olimond BB”? Please provide the detailed information.
In the simple summary we added an explanation “…based on products of fermented yeasts (Saccharomyces cerevisiae)”
2. Line 100, 1) the table tile need to be more specified, such as the name of the pellet; 2) please add the full name explanation of PLA and OLI in the table note; 3) what is the total amount for each pellet?
The table was enhanced and the table note was changed. The total amount (in g/pellet) was added to the table
3. Too much details for the material and methods. Please reduce them by cite the reference.
We significantly reduced the information provided in 2.3 and 2.4 (we combined 2.3. and 2.4 into 2.3) and provide a reference
4. Line 201, 1) please do not use the bond style for D40 and D41. Please check the similar issues throughout the paper; 2) why not use the mean ± SD to express the results?
1) We changed the bold style into normal style in L210 (Table 2) and L245 (Table 4). 2) SD gives an indication how close samples are to each other while SEM gives an impression about the accuracy of the mean.
5. Line 218, please correct “P-value” to “P-value”.
Done
6. It is not the right way to express the “Olimond BB (OLI) or Placebo (PLA)” in the table note. It should be appeared as “OLI, Olimond BB; PLA, Placebo”.
Done
7. Line 233, please correct “(P = < 0.005)” to “P ≤ 0.005”.
Done (this was in L 223)
8. Line 229, please correct “(OLI, n =6; PLA, n =5)” to “(OLI, n = 6; PLA, n = 5)”.
Done
9. Line 2447, please correct “vaules ± standard deviation” to “means ± SEM”, which should consistent with the previous presentation.
Done
10. Please carefully check the reference and make them follow the right style of the journal.
We thoroughly revised the references according to the required style.
Round 2
Reviewer 2 Report
Thanks for your clarification. No further comment.
Author Response
Thank you very much for your final approval